# *n*-3 Polyunsaturated Fatty Acids Decrease Long-Term Diabetic Risk of Offspring of Gestational Diabetes Rats by Postponing Shortening of Hepatic Telomeres and Modulating Liver Metabolism

**DOI:** 10.3390/nu11071699

**Published:** 2019-07-23

**Authors:** Jinlong Gao, Hailong Xiao, Jiaomei Li, Xiaofei Guo, Wenwen Cai, Duo Li

**Affiliations:** 1Department of Food Science and Nutrition, Zhejiang University, 866 Yuhangtang Road, Hangzhou 310058, China; 2Hangzhou Institute for Food and Drug Control, 198 Yonghua Street, Hangzhou 310022, China; 3Institute of Nutrition and Health, Qingdao University, 308 Ningxia Road, Qingdao 266071, China

**Keywords:** gestational diabetes mellitus, offspring, long-term effect, n-3 polyunsaturated fatty acid, diabetes risk, liver, oxidative stress, inflammation, telomere length, metabolomics

## Abstract

The long-term influence of gestational diabetes mellitus (GDM) on offspring and the effect of omega-3 polyunsaturated fatty acids (*n*-3 PUFA) on GDM offspring are poorly understood. We studied the long-term diabetic risk in GDM offspring and evaluated the effect of *n*-3 PUFA intervention. Healthy offspring rats were fed standard diet (soybean oil) after weaning. GDM offspring were divided into three groups: GDM offspring (soybean oil), *n*-3 PUFA adequate offspring (fish oil), and *n*-3 PUFA deficient offspring (safflower oil), fed up to 11 months old. The diabetic risk of GDM offspring gradually increased from no change at weaning to obvious impaired glucose and insulin tolerance at 11 months old. *n*-3 PUFA decreased oxidative stress and inflammation in the liver of older GDM offspring. There was a differential effect of *n*-3 PUFA and *n*-6 PUFA on hepatic telomere length in GDM offspring. Non-targeted metabolomics showed that *n*-3 PUFA played a modulating role in the liver, in which numerous metabolites and metabolic pathways were altered when GDM offspring grew to old age. Many metabolites were related to diabetes risk, such as α-linolenic acid, palmitic acid, ceramide, oxaloacetic acid, tocotrienol, tetrahydro-11-deoxycortisol, andniacinamide. In summary, GDM offspring exhibited obvious diabetes risk at old age, whereas *n*-3 PUFA decreased this risk.

## 1. Introduction

Gestational diabetes mellitus (GDM) increases the offspring’s risk of developing obesity, diabetes, cardiovascular disease, and other diseases [1]. The long-term impact of GDM on offspring, however, is poorly understood. The risk and mechanisms of developing diabetes during GDM offspring growth, especially as they grow to an old age, are not well known. N-3 polyunsaturated fatty acids (PUFA) play a role in preventing cardiovascular disease, hypertension, cancer, and cognitive decline [2]. Nevertheless, the relationship between *n*-3 PUFA and diabetes risk varies from study to study, even with opposite conclusions [3,4]. Furthermore, it remains unclear whether *n*-3 PUFA can decrease diabetic risk for GDM offspring. Several studies have suggested possible mechanismsfor GDM offspringto develop diabetes, including an abnormal insulin signaling pathway, lower antioxidant capacity, lipid metabolism disorder, and epigenetic alterations [5,6,7,8]. However, other mechanisms need to be studied to prevent the diabetic occurrence in GDM offspring.

Telomeres, which are termini of chromosomes, contain specific DNA sequences. Telomeres are essential for chromosomal integrity. The length of telomeres shortens with age [9]. Accelerated telomere shortening is caused by some adverse factors such as oxidative stress and inflammation [10]. Shortening of telomere length is a biomarker of aging, and it is linked to diseases risk [11]. Metabolomics plays a role in studying metabolic changes. Several studies based on metabolomics have focused on GDM mothers, investigating the correlation between abnormal metabolites and GDM or finding biomarkers that can predict GDM risk [12,13,14]. Only a few studies have analyzed metabolic changes of GDM offspring at birth and infancy [15,16]. Nevertheless, long-term effects of GDM on metabolic alterations in offspring are not well known. 

The liver is the main organ for the regulation of blood glucose [17]. Theory of developmental origins of health and disease (DOHaD theory) and fetal programming hypothesis reveal that the risk of adult diseases later in life is associated with adverse intrauterine conditions [18,19]. GDM is a typical adverse intrauterine condition in which placental transport of nutrients to the fetus is altered and some adverse factors such as oxidative stress and inflammation cause damage to the fetal liver and other organs [20]. First, aging is an inevitable process. This lets us question whether GDM can cause hepatic aging of offspring later in life and whether the length of hepatic telomeres in GDM offspring will be obviously shortened at old age. Furthermore, the effect of *n*-3 PUFA and *n*-6 PUFA on the length of hepatic telomeres in GDM offspring remains unclear. Second, the liver is the center of metabolism. This allow us to consider whether GDM can lead to long-term metabolic changes of the liver of offspring, increasing the risk of developing diabetes at old age. Moreover, the long-term effect of *n*-3 PUFA and *n*-6 PUFA on hepatic metabolism of GDM offspring has not been studied. 

The aim of this study was to investigate the risk of developing diabetes during GDM offspring rats’ growth, particularly when they grow to 11 months old. Meanwhile, the effect of *n*-3 PUFA on diabetes risk of GDM offspring was studied. For the first time, this study discovered the differential effect of *n*-3 PUFA and *n*-6 PUFA on the length of hepatic telomeres in GDM offspring. Moreover, metabolomics analysis revealed obvious metabolic alterations of the liver of GDM offspring at an old age. *n*-3 PUFA played an important role in modulating the metabolism of the liver of GDM offspring. 

## 2. Materials and Methods

### 2.1. Animal Model and Offspring Diet Intervention

The animal protocols were approved by Zhejiang University Animal Center and Ethic Committee. The protocol of the study is shown in Figure 1. Forty-four female 10-week-old virgin Wistar female rats and 22 male adult rats were obtained and fed purified AIN-93 diet. The diet composition was as follows (g/kg): starch, 397; maltodextrin, 132; casein, 200; sucrose, 100; soybean oil, 70; fiber, 50; mineral mix, 35; vitamin mix 10; L-cystine, 3; choline bitartrate, 2.5; Tert-butylhydroquinone, 0.14. After one week of acclimation, female rats were randomly divided into two groups: healthy maternal group (*n* = 10) and GDM group (*n* = 34). Female rats, two per cage, mated with male rats overnight by ratio of 2:1. Day 0 of gestation was confirmed by presence of sperm in vaginal smears. On the fifth day of gestation, diabetes was induced at the fasting state by a single intraperitoneal injection of streptozotocin (STZ; Sigma, St. Louis, MO, USA) in 0.1 mmol/L citrate buffer (pH 4.5) at a dose of 30 mg/kg bodyweight. Healthy pregnant rats received an equal volume of citrate buffer alone. Blood glucose was measured on D12 and D17 of pregnancy via tail by One Touch Glucometer (Johnson & Johnson, USA). Diabetic pregnant rats, whose blood glucose ranged between 15 and 19 mmol/L both in two occasions, were included. Pregnant rats were housed individually and allowed to deliver spontaneously. Each litter size was reduced to six pups (three male and three female) to assure uniformity. The pups were weaned at three weeks of age. Respective 10 pups from healthy and GDM maternal rats were randomly selected to be sacrificed after weaning. The remaining offspring from healthy rats were fed an AIN-93 diet (7% soybean oil) to 11 months old as a control offspring group (Con group). Offspring from GDM rats were randomly divided into three groups: GDM offspring (GDM group, fed 7% soybean oil diet), GDM offspring with adequate *n*-3 PUFA diet (n-3 Adq-GDM group), and GDM offspring with deficient *n*-3 PUFA diet (n-3 Def-GDM group or high *n*-6 PUFA group). These rats were fed their respective diet until they were 11 months old. In the n-3 Adq-GDM group, 4% of the 7% of the soybean oil in the AIN-93 diet was replaced with fish oil containing 60% *n*-3 PUFA (50% docosahexaenoic acid (DHA), 10% eicosapentaenoic acid (EPA)) and the ratio of *n*-3 PUFA to *n*-6 PUFA was increased from 1:8 to 1.5:1. In then-3 Def-GDM group, 7% soybean oil was completely substituted by safflower oil, rendering an obvious decreased ratio of n-3/*n*-6 PUFA to 1:95. All purified diets were produced once a week (Medicience Ltd., Jiangsu, China) and kept at −40 °C. Diets were replaced every day and uneaten food was discarded. All rats were maintained with a 12h light/dark cycle at 22 °C and had free access to food and water. Body weight and food intake were measured weekly. After overnight fasting, the animals were sacrificed at 11 months of age by intraperitoneal administration of chloral hydrate (350 mg/kg bodyweight). Blood was collected from the abdominal aorta and was centrifuged at 3000 r/min for 10 min, 4 °C. The livers were dissected and immediately placed into liquid nitrogen. They were subsequently stored at −80 °C for analyses. 

### 2.2. Determination of Serum Factors

The levels of serum glucose, triglycerides, total cholesterol, and high-density lipoprotein (HDL) were determined by biochemical analyzer (Roche, Basel, Switzerland). Insulin was determined in duplicate using ELISA assays (Nanjing Jiancheng Bioengineering Institute, Nanjing, China) according to manufacturer’s instructions. 

### 2.3. Glucose and Insulin Tolerance Tests

Offspring glucose tolerance test (GTT) and insulin tolerance test (ITT) were performed at 3 weeks, 3 months, and 11 months (a week before being sacrificed). For GTT, glucose (2 g/kg) was intraperitoneally injected after 12 h overnight fasting. Blood glucose was measured via tail by glucometer at 0, 15, 30, 60, and 120 min. For ITT, insulin (0.5 unit/kg) was intraperitoneally injected after 4 h fasting. Blood glucose was measured at 0, 30, 60, 90, and 120 min.

### 2.4. Measurement of Oxidative Stress

Liver tissues were homogenized using a multifunctional homogenizer. The protein concentrations were measured by colorimetric assay. The activity of superoxide dismutase (SOD), glutathione peroxidase (GSH-Px), and catalase (CAT) as well as the levels of malondialdehyde (MDA) and glutathione (GSH) were determined using commercial kits following manufacturer’s instructions (Nanjing Jiancheng Biotechnology Institute, Nanjing, China) [21].

### 2.5. Enzyme-Linked Immunosorbent Assay for Inflammatory Factors

Commercially available ELISA kits (Nanjing Jiancheng Biotechnology Institute, Nanjing, China) were used to determine TNF-α, IL-1β, IL-6, and IL-10 levels in liver tissue of offspring rats in accordance with the manufacturer’s instruction [22]. 

### 2.6. Telomere Length Measurement

DNA from the liver of offspring rats was isolated using DNA extraction kit (Tiangen, Beijing, China). Relative telomere lengths were measured by qPCR method with modifications [23]. In brief, DNA samples were used for telomeres and single-copy gene (36B4) qPCR, which was amplified by Power SYBR Green in 20 µL total volume. The reaction was performed on a Light Cycler 480 II with the following conditons: 95 °C for 3 min, followed by 45 cycles of 95 °C for 3 s and 60 °C for 30 s, followed by a dissociation stage to monitor amplification specificity. The telomere length was calculated based on 2^−ΔΔCt^. The tested genes and primer sequence were (5′3′): Tel1: GGTTTTTGAGGGTGAGGGTGAGGGTGAGGGTGAGGGT; Tel2: TCCCGACTATCCCTATCCCTATCCCTATCCCTATCCCTA; 36B4 Fw: CAGCAAGTGGGAAGGTGTAATCC; 36B4 Rv: CCCATTCTATCATCAACGGGTACAA.

### 2.7. Metabolomics Analysis

#### 2.7.1. Sample Preparation

Non-targeted metabolomics analysis on liver tissue of offspring rats was performed. Twenty micrograms of each liver sample were weighed out into 2 mL EP tubes and homogenized with 800 µL of prechilled 80% methanol by an automatic tissue homogenizer (Prima, London, UK). Homogenization cycles were for 40 s followed by cooling on dry ice and further 40 s homogenization and cooling. The mixtures were stored at −20 °C for 1 h, and then centrifuged at 14,000 *g* for 15 min at 4 °C. The collected supernatants were filtered by a 0.22 um filtering membrane and transferred to auto sample vials (Waters Corporation, Milford, MA, USA) for analysis. 

#### 2.7.2. High Performance Lipid Chromatography Coupled with Quadrupole-Time of Flight Mass Spectrometry (HPLC-QTOF-MS) Analysis

HPLC-QTOF-MS analysis was performed on Waters XEVO G2 Q-TOF mass spectrometer system (Waters Corporation, USA). Separation was performed at 50 °C and 0.3 mL/min using an Acquity UPLC BEH C18 (Waters Corporation, USA) column (2.1 × 100 mm, 1.7 um). The injection volume was 2 µL and sample temperature was 4 °C. A mixture of 0.1% formic acid in distilled water (mobile phase A) and 0.1% formic acid in acetonitrile (mobile phase B) was used. The initial eluent was 20% B for 0–3.0 min, followed by a linear gradient from 20% to 50% B for 7.0 min. Mobile phase B was then increased from 50% to 90% over 7 min and subsequently to 100% for 3.0 min. Mobile phase B was subsequently returned to initial conditions to re-equilibrate for 4.0 min. MS acquisition was performed in positive ionization mode with a mass-to-charge (m/z) of 50–1200. Source conditions were as follows: source temperature, 120 °C; desolvation temperature, 300 °C; cone gas flow, 50 L/h; desolvation gas flow, 800 L/h.

#### 2.7.3. Data Processing and Metabolites Identification

LC/MC raw data were processed by the software Progenesis QI (Waters Corporation, USA) using the following parameters, retention time (RT) range 0.5–20 min, mass range 50–1000 Da, and mass tolerance 0.01 Da. Isotopic peaks were excluded for analysis. The noise elimination level was at 10.00, the minimum intensity was set to 15% of base peak intensity, and RT tolerance was at 0.01 min. Metabolites identification was performed in Progenesis QI based on appropriate standards and mass fragmentation (MS/MS analysis). The available online database (HMDB) was used to identify the name and chemical structure of metabolites by molecular formula information. The file was obtained with three-dimensional data sets, including peak RT, peak intensities, and m/z. The resulting matrix was further reduced through removing peaks with missing value in more than 60% samples. 

### 2.8. Statistical Analysis

Data are expressed as the mean ± standard deviation (SD). Student’s *t*-tests were used to determine significant differences between two groups. Significant differences between four groups were determined by a one-way ANOVA followed by an LSD analysis using SPSS 20.0. Differences were considered statistically significant if the *p*-value was less than 0.05. For statistical analysis of data from metabolomics, the data that were pre-processed by the software Progenesis QI were transferred into SIMCA-P+14.0 (Umetircs, Umea, Sweden) to conduct multivariate statistical analysis. Principle component analysis (PCA) and orthogonal partial least-squares-discriminant analysis (OPLS-DA) were carried out to visualize the metabolic alterations among groups. Metabolites with *p* < 0.05 and variable importance on projection (VIP) > 1 were considered relevant for group discrimination.

## 3. Results

### 3.1. Serum Biochemical Index of GDM Offspring and Effect of *n*-3 PUFA

As shown in Table 1, at weaning, there was no significant difference in fasting and postprandial glucose between Con and GDM offspring. The level of insulin of GDM offspring at weaning was significantly decreased in the fasting state (*p* < 0.05), compared to that of Con offspring. However, in the postprandial state, insulin of GDM offspring was significantly increased (*p* < 0.01), indicating early dysfunction of β cells caused by severe maternal hyperglycemia. No significant changes were observed in triglyceride and total cholesterol between the Con offspring and GDM offspring at weaning. As shown in Table 2, when GDM offspring grew to an age of 11 months, the fasting insulin level was still significantly decreased (*p* < 0.01) with a higher fasting glucose level (*p* > 0.05, not significant) compared to that of Con offspring. N-3 Adq-GDM offspring exhibited significantly increased insulin (*p* < 0.01) compared to that of GDM offspring. No improvement of insulin was observed in n-3 Def-GDM group. Supplementation of *n*-3 PUFA significantly decreased triglyceride and total cholesterol in serum of GDM offspring (*p* < 0.05 and *p* < 0.01). Meanwhile, *n*-3 PUFA also significantly decreased the ratio of total cholesterol to high-density lipoprotein (TC/HDL), which was a more meaningful result than the value of HDL alone.

### 3.2. Glucose and Insulin Tolerance Test during Growth of GDM Offspring and Effect of *n*-3 PUFA

GTT and ITT were performed to evaluate the risk of developing diabetes during growth of GDM offspring. The diabetic risk in GDM offspring gradually increased with age. At weaning, no obvious GTT and ITT differences were observed (Figure 2A,D). At 3 months of age, GDM offspring exhibited a higher blood glucose level only at 15 min in GTT, compared to that of Con offspring (Figure 2B). In ITT, the blood glucose level of GDM offspring was higher only at 30 min and 60 min (Figure 2E). Surprisingly, when GDM offspring grew to 11 months of age, the GTT was obviously impaired. Furthermore, GDM offspring receiving an*n*-3 PUFA deficient diet exhibited an even more obvious impairment in GTT. GDM offspring with adequate *n*-3 PUFA diet; however, showed a significant decrease in the GTT (Figure 2C). Likewise, *n*-3 PUFA supplementation also decreased the ITT in GDM offspring at 11 months of age, whereas the worst ITT was observed in *n*-3 PUFA deficient offspring (Figure 2F). 

### 3.3. Effect of *n*-3 PUFA on Oxidative Stress of the Liver of GDM Offspring

Oxidative stress in the liver was measured (Figure 3). The glutathione (GSH) level, the activity of superoxide dismutase (SOD) and glutathione peroxidase (GSH-Px) were decreased in the liver of GDM offspring (*p* < 0.01, *p* < 0.01 and *p* < 0.05, respectively). *n*-3 PUFA significantly enhanced the activity of SOD and catalase (CAT) in the liver of GDM offspring (*p* < 0.01). No improvement was observed in n-3 Def-GDM offspring and there was a differential effect on activity of SOD and GSH-Px between n-3 Adq-GDM offspring and n-3 Def-GDM offspring (*p* < 0.01 and *p* < 0.05). 

### 3.4. Effect of *n*-3 PUFA on Inflammatory Factors in the Liver of GDM Offspring

As shown in Figure 4, the levels of hepatic TNF-α and IL-1β were increased in GDM offspring (*p* < 0.05). The level of IL-10, an anti-inflammatory cytokine, was decreased in the liver of GDM offspring (*p* < 0.01). *n*-3 PUFA decreased IL-1β and increased IL-10 levels (*p* < 0.05). N-3 Def-GDM offspring exhibited the highest level of TNF-α and IL-1β in all groups. There was a significant opposite trend in IL-1β (*p* < 0.01) and IL-10 (*p* < 0.01) between the n-3 Adq-GDM group and n-3 Def-GDM group. No significant difference was observed in IL-6 levels. 

### 3.5. Influence of GDM on Telomere Length of the Liver of Offspring and the Effect of *n*-3 PUFA and *n*-6 PUFA on Telomere Length

At weaning, there was no significant difference in length of the hepatic telomeres between Con offspring and GDM offspring (Figure 5A). However, as shown in Figure 5B, telomere length of the liver of GDM offspring at 11 months of age was nearly significantly decreased, compared with that of Con group (*p* = 0.058). Furthermore, telomere length of the liver of n-3 Def-GDM offspring was significantly shortened, compared with that of Con offspring (*p* < 0.05). Additionally, n-3 Def-GDM offspring exhibited the shortest hepatic telomere length of all four groups. The telomere length of the liver of n-3 Adq-GDM offspring was nearly improved by supplementation with *n*-3 PUFA, compared with that of GDM offspring (not significant, nearly significant, *p* = 0.081). It is noticeable that there is an obvious opposite trend of hepatic telomere length between n-3 Adq-GDM offspring and n-3 Def-GDM offspring (*p* < 0.05), suggesting a differential effect of *n*-3 PUFA and *n*-6 PUFA on the telomere length of the liver of GDM offspring.

### 3.6. Metabolomics Analysis of the Liver of GDM Offspring at Old Age and the Modulating Effect of *n*-3 PUFA on the Liver of GDM Offspring

Samples of the liver were clustered within their respective groups in a PCA model and further clustered in OPLS-DA model (Appendix A). Pairwise OPLS-DA modeling was conducted to assess data quality and to screen differential metabolites. The value of R^2^Y and Q^2^Y were high, indicating a good model quality (Appendix A). A permutation test was performed to further confirm the model’s validity (Appendix A). For maximum screening of all significantly altered metabolites, metabolites were selected based only on variable importance in projection (VIP) values of greater than 1 and *p* < 0.05, without further screening by fold change. 

Metabolomics analysis revealed obvious metabolic alterations of the liver when GDM offspring grew to 11 months of age. Some representative metabolites were closely related to oxidative stress, inflammation and risk of developing diabetes (Table 3). A total of 73 metabolites were significantly altered in the liver of GDM offspring, with 38 increased metabolites and 35 decreased ones (Appendix A and Table 3). Twenty-seven metabolites of the altered 73 metabolites were improved in the n-3 Adq-GDM group. Only seven metabolites in the liver of n-3 Def-GDM group were improved but changing trend of other 21 metabolites were aggravated. Obviously, more metabolites were improved in the n-3 Adq-GDM group rather than in the high *n*-6 PUFA group (n-3 Def-GDM group). *n*-3 PUFA played an important role in modulating metabolism of the liver of GDM offspring. 

Meanwhile, the enrichment analysis of multiple metabolic pathways was performed (Appendix A). Some of these altered metabolic pathways were closely related to glucose and lipid metabolism, and diabetes risk, including citric acid cycle, pyruvate metabolism, glycolysis, gluconeogenesis, linoleic acid, and linolenic acid metabolism. 

## 4. Discussion

Long-term effects of GDM on offspring are poorly understood. We studied the risk of developing diabetes during growth of GDM offspring, particularly as they grew to an old age. To study the long-term diabetic risk of GDM offspring, we evaluated two hypotheses. The first is that GDM can lead to aging of the liver of offspring, which increases the risk of developing diabetes. To evaluate this hypothesis, we measured oxidative stress, inflammation, and telomere length. The second is that GDM can cause long-term metabolic changes in the liver of offspring. If so, we evaluate whether such changes are associated with diabetic risk. The discussion is primarily based on these two hypotheses (Figure 6). Meanwhile, the effect of *n*-3 PUFA on the diabetic risk in GDM offspring was also studied. Oil replacement is a common method to study the effect of *n*-3 PUFA. According to data determined in our laboratory, the ratio of *n*-3 PUFA to *n*-6 PUFA in diets of the Con group, n-3 Adq-GDM group, and n-3 Def-GDM group (high *n*-6 PUFA group) were 1:8, 1.5:1, and 1:95, respectively. Metabolomics performed by UPLC-QTOF-MS detected changes of DHA in the liver. The mean abundance value of DHA in the Con, GDM, n-3 Adq-GDM, and n-3 Def-GDM groups were 3429, 4422, 15966, and 1356, respectively (Con vs. GDM, not significant; n-3 Adq-GDM group vs. Con, GDM and n-3 Def group, *p* < 0.01; n-3 Def-GDM group vs. Con, GDM and n-3 Adq-GDM group, *p* < 0.01), showing that fed n-3 fatty acids from fish oil diet were successfully incorporated into n-3 Adq-GDM offspring.

Low birth weight is associated with adult diseases including type 2 diabetes [24]. In the present study, the birth weight of GDM offspring was significantly decreased (Appendix A), suggesting their increased risk of developing diabetes later in life. GDM offspring exhibited a lifelong growth restriction, which was improved in the n-3 Adq-GDM group, but not in the n-3 Def-GDM group, indicating a protective effect of *n*-3 PUFA on GDM offspring. Moreover, triglyceride levels were increased in the liver of GDM offspring at weaning and even progressed up to 11 months of age (Appendix A). Hepatic triglycerides have been associated with insulin resistance and diabetes risk [25]. Troy Pereira found increased TG in the liver of “obese” offspring from “obese” GDM rats [7]. However, we found that hepatic TG was increased in “lean” offspring from “lean” GDM rats induced by STZ. This result indicates that regardless of the GDM model, intrauterine hyperglycemia can lead to hepatic lipid metabolism disorders in offspring, and these disorders can even continue to old age, which increases the risk of developing diabetes. *n*-3 PUFA decreased the levels of triglyceride and cholesterol in the liver of GDM offspring, suggesting a protective effect on diabetic risk in GDM offspring.

Studies have shown that the risk of developing diabetes in GDM offspring was different. A ref et al. found abnormal fasting glucose in offspring rats at postnatal weeks one and two [26]. In Ding’s study, GTT was impaired in 8-weeks-old offspring mice [27]. Kamel et al. found impaired GTT in offspring rats at 15 weeks of age [28]. Population studies found that the diabetic risk in GDM offspring was increased at childhood, puberty, and adulthood [29,30,31]. However, longer-term diabetic risk, even after the offspring have grown to an old age, has been poorly studied. Furthermore, the relationship between *n*-3 PUFA and diabetes risk varies from study to study, even with opposite conclusions [3,4]. Our results showed that the diabetes risk of GDM offspring gradually increased, with no obvious change in the GTT and ITT at weaning to slight changes at 3 months old. Surprisingly, GDM offspring exhibited obvious impairment in the GTT and ITT at 11 months. *n*-3 PUFA improved GTT and ITT of GDM offspring at 11 months, whereas more obviously impaired GTT and ITT were observed in n-3 Def-GDM offspring, indicating the beneficial effect of *n*-3 PUFA on the diabetic risk in GDM offspring. 

Our present study showed that the liver of GDM offspring exhibited oxidative stress and inflammation at 11 months of age, which were important factors in inducing diabetic risk. SOD is an important defense enzyme catalyzing dismutation of superoxide radicals [32]. The decreased activity of SOD in the liver of GDM offspring lowers cellular capacity to scavenge free radicals, which reduces hepatic cell protection from oxidant exposure, resulting in diabetic risk. In the diabetic state, glucose is preferentially utilized in the polyol pathway, which consumes nicotinamide adenine dinucleotide phosphate (NADPH) necessary for GSH regeneration. Thus, there is a link between diabetes and GSH depletion [32]. Furthermore, low levels of GSH could directly decrease activity of GSH-Px, which is a main antioxidant enzyme that protects cells against hydroperoxides caused by reactive ROS [33]. Therefore, decreased GSH levels and GSH-Px activity in the liver of GDM offspring were factors for developing diabetes. *n*-3 PUFA increased the activity of SOD and it also increased the activity of CAT, which reduces intracellular ROS and inhibits oxidative damage [34], preserving its protective role in diabetes development. TNF-α is one of the most important pro-inflammatory mediators that is critically involved in the pathogenesis of diabetes by inducing tissue-specific inflammation via activation of various pathways such as NF-κB. TNF-α decreases GLUT4 and phosphorylation of IRS-1 [35]. IL-1β is also a key contributor to the pathogenesis of diabetes. It contributes to insulin resistance in the liver [36]. TNF-α and IL-1β were increased in the liver of GDM offspring, whereas IL-10, an anti-inflammatory cytokine, was decreased in GDM offspring, indicating an increased risk of developing diabetes. There was a significantly opposite trend in TNF-α, IL-1β, and IL-10 between the n-3 Adq-GDM group and n-3 Def-GDM group, which could lead to differential diabetes risk. 

Furthermore, oxidative stress and inflammation are important factors leading to telomere shortening [37,38]. The length of telomere in the liver of GDM offspring was nearly significantly shortened at 11 months of age (*p* = 0.058), indicating potential aging of the liver. A population study found that telomere length was shortened in leucocytes of the fetus from mothers with GDM, suggesting the possibility of aging of GDM offspring in the future [39]. However, other studies showed that no significant telomere shortening was observed in neonatal offspring and adult offspring from mothers with diabetes [40,41]. We observed a nearly shortened telomere length in the liver when GDM offspring grew to 11 months of age. It can be speculated that if the rats continued to age past 11 months, the length of telomeres in the liver of GDM offspring would be significantly shortened and hepatic aging may be more obviously. Because the liver is the main organ for the regulation of blood glucose, hepatic aging of GDM offspring may reduce the liver’s ability to regulate blood glucose, resulting in risk of developing diabetes. We observed an opposite result on the length of the hepatic telomeres in the liver between n-3 Adq-GDM offspring and n-3 Def-GDM offspring, suggesting that *n*-3 PUFA and *n*-6 PUFA played a differential role in telomere length. GDM offspring with a*n*-3 PUFA deficient diet exhibited the shortest telomere length, whereas *n*-3 PUFA postponed shortening of the telomere.

Our study is the first to investigate the long-term influence of GDM on metabolic changes in the liver of offspring. Surprisingly, many metabolites were altered in the liver when GDM offspring grew to 11 months. Some typical metabolites were closely related to diabetes risk. A particularly novel finding was that ceramide was increased in the liver of GDM offspring. Ceramide is a typical biomarker for diabetes risk. Excessive accumulation of ceramide in the liver impairs insulin signaling and contributes to diabetic development [42]. Moreover, hexadecenoic acid was increased in the liver of GDM offspring, which could induce lipotoxicity and insulin resistance, increasing the risk of developing type 2 diabetes [43]. It is worth noting that oxaloacetic acid was also decreased in the liver of GDM offspring, which directly affects the citric acid cycle and further affects glucose and lipid metabolism. A decrease inoxaloacetic acidalso might indicate abnormal gluconeogenesis [44]. Tetrahydro-11-deoxycortisol was increased in the liver of GDM offspring, which could further increase the level of cortisol. Cortisolcan affect insulin production and glucose metabolism and it can cause insulin resistance [45,46]. α-linolenic acid (ALA) can improve tissue’s insulin resistance [47,48]. ALA is also associated with the lowest risk of type 2 diabetes [49]. ALA was decreased in the liver of GDM offspring, which might be related to the risk of developing diabetes. Niacinamide can be used to prevent the development of diabetes by reducing β-cell apoptosis [50,51,52]. Niacinamide was decreased in the liver of GDM offspring, which might be a risk factor for the development of diabetes. Meanwhile, it is worth noting that some altered metabolites were not only related to diabetic risk, but also were associated with oxidative stress, inflammation, aging, and hepatic function, such as 9’-carboxy-gamma-tocotrienol [53,54], α-linolenic acid, hexadecenoic acid [55], ceramide [56], niacinamide [57,58], and phenylethylamine [59,60]. These metabolites were consistent with the results of the enhanced oxidative stress and inflammation, and shortening of telomeres in the liver of GDM offspring at an old age. Thus, the second hypothesis based on metabolomics was not isolated, but rather linked with the first hypothesis.

All metabolites altered in the liver of GDM offspring involve various metabolic pathways, some of which were closely related to glucose and lipid metabolism, as well as diabetic risk, such as citric acid cycle, pyruvate metabolism, glycolysis, gluconeogenesis, linoleic acid, and linolenic acid metabolism (Appendix A). Our resultsindicate that hyperglycemia uterine conditions, which the fetus experienced early in life, did cause long-term metabolic disorders to the liver of offspring later in life. The liver is the center of metabolism and a key site for metabolic disorders in patients with type 2 diabetes. Compared with the possibility of results explained by transcriptome and proteomics, metabolomics can explain what ultimately happened. Thus, the changes of these metabolites and metabolic pathways demonstrated an important molecular basis and mechanism for the increased risk of GDM offspring at old age. Our results also provided evidence and a molecular basis for the theory of developmental origins of health and disease. *n*-3 PUFA played a role in improving the overall metabolism of the liver of GDM offspring. Changing trends of some metabolites were even aggravated in the high *n*-6 PUFA group. For the first time, the differential effects of *n*-3 PUFA and *n*-6 PUFA on hepatic metabolism of GDM offspring were compared in this study.

It is worth noting that the overall physical condition of GDM offspring was very poor. Limbs of GDM offspring were thin at weaning and 11 months old (Figure 7). Some of the n-3Def-GDM offspring even could not stand properly, which may be partly because of muscle weakness. This observed muscle weakness implies that these animals are unable to perform adequate exercise, which in turn may also enhance insulin resistance and predispose them to develop diabetes. It is possible that the muscle tissues of these animals have fewer number of mitochondria and that actin-myosin and filaments may be improperly organized. The fatty acid pattern of muscle tissue may be different. Whereas, the performance of n-3 Adq-GDM offspring was improved. This could be partly because *n*-3 PUFA improved insulin resistance in muscle tissue in GDM offspring, which deserves further investigation.

In summary, GDM offspring exhibited an obvious increased risk for diabetes when they grew to an old age, whereas *n*-3 PUFA decreased this risk. To the best of our knowledge, this is the first study to investigate the long-term influence of GDM on metabolism of the liver of offspring and the modulating effect of *n*-3 PUFA on hepatic metabolism. Surprisingly, the differential effect of *n*-3 PUFA and *n*-6 PUFA on the length of hepatic telomeres in GDM offspring was discovered in this study. Finally, our results also provide evidence and a molecular basis for the DOHaD theory and fetal programming hypothesis.

## Figures and Tables

**Figure 1 nutrients-11-01699-f001:**
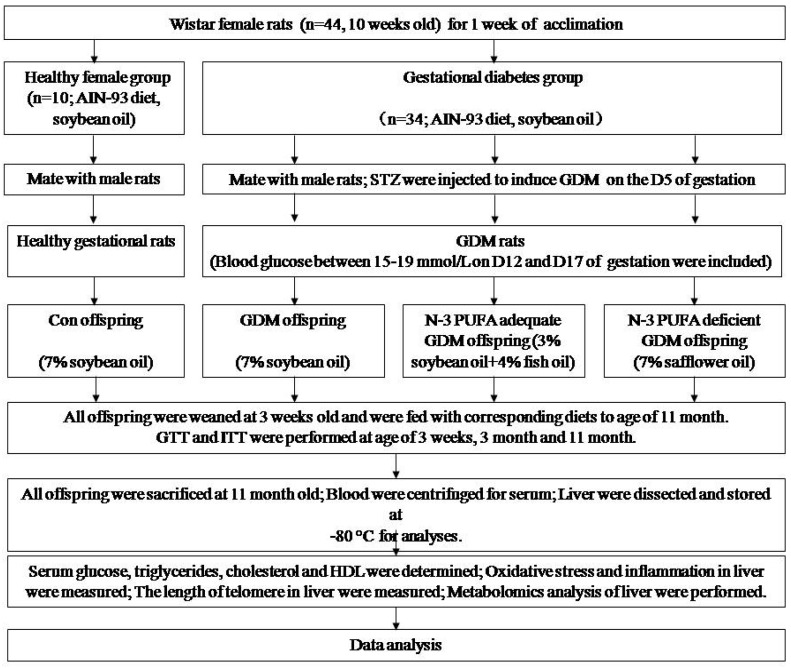
Flow diagram showing the protocol of the study.

**Figure 2 nutrients-11-01699-f002:**
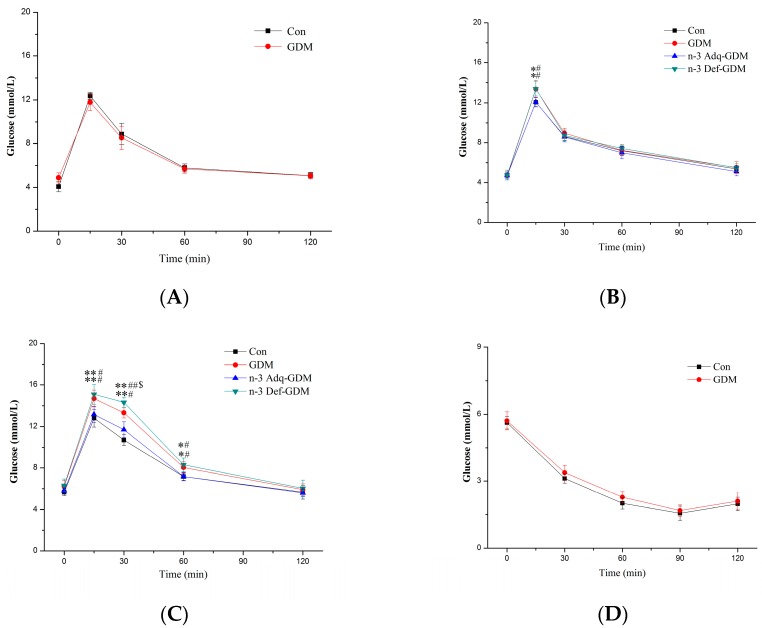
Glucose tolerance test (GTT) and insulin tolerance test (ITT) in growing gestational diabetes mellitus (GDM) offspring and the effect of n-3 polyunsaturated fatty acids (*n*-3 PUFA). (**A**) GTT at weaning; (**B**) GTT at 3 months; (**C**) GTT at 11 months; (**D**) ITT at weaning; (**E**) ITT at 3 months; (**F**) ITT at 11 months. Values are the mean ± SD. *n* = 8 rats/group. * *p* < 0.05, ** *p* < 0.01, vs. Control offspring (Con); ^#^
*p* < 0.05, ^##^
*p* < 0.01, vs. n-3 Adequate-GDM offspring (n-3 Adq-GDM); ^$^
*p* < 0.05, n-3 Deficient-GDM offspring (n-3 Def-GDM) vs. GDM offspring (GDM).

**Figure 3 nutrients-11-01699-f003:**
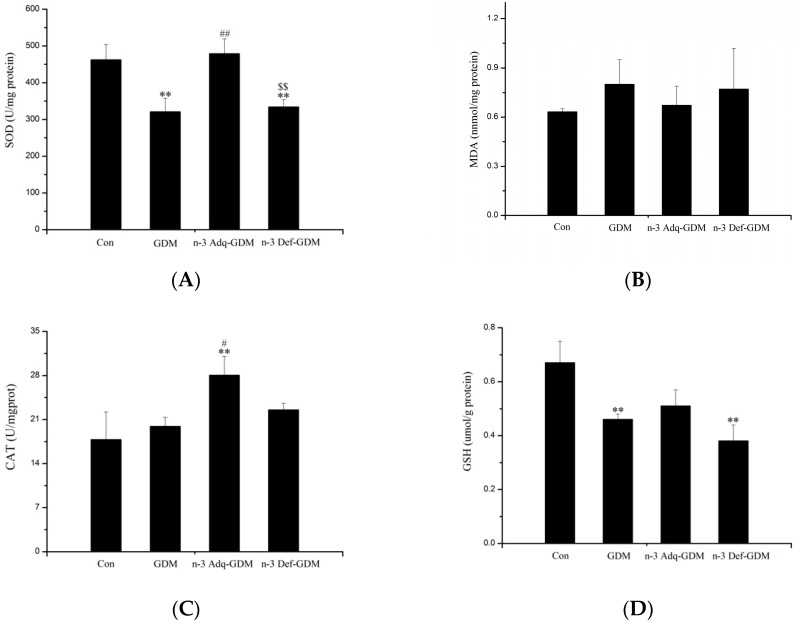
Oxidative stress in the liver of gestational diabetes mellitus (GDM) offspring at 11 months of age and the effect of n-3 polyunsaturated fatty acids (*n*-3 PUFA) on oxidative stress. (**A**)The activity of SOD, (**B**) MDA, (**C**) The activity of CAT, (**D**) GSH, (**E**) The activity of GSH-Px. Bars are mean ± SD. *n* = 8 rats/group. * *p* < 0.05, ** *p* < 0.01, vs. Control offspring (Con); ^#^
*p* < 0.05, ^##^
*p* < 0.01, vs. GDM offspring (GDM); ^$^
*p* < 0.05, ^$$^
*p* < 0.01, vs. n-3 Adequate-GDM offspring (n-3 Adq-GDM).

**Figure 4 nutrients-11-01699-f004:**
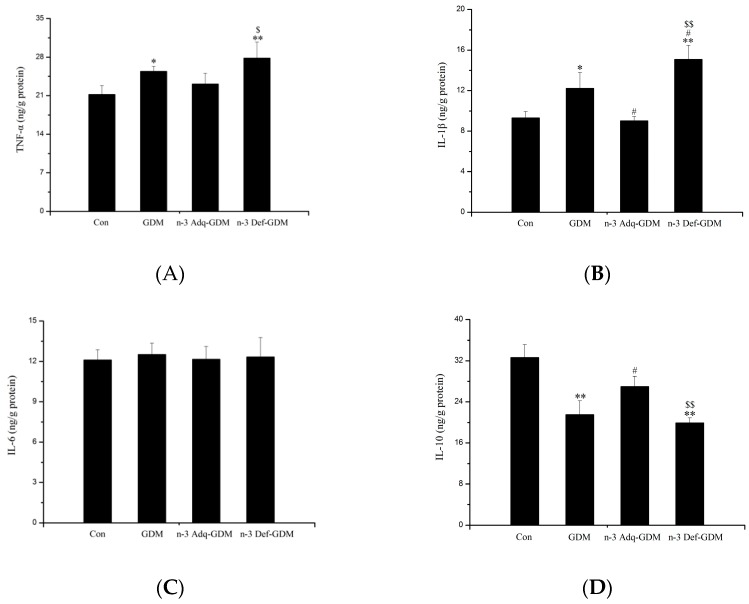
Inflammatory biomarkers in the liver of gestational diabetes mellitus (GDM) offspring at 11 months of age and the effect of n-3 polyunsaturated fatty acids (*n*-3 PUFA) on inflammatory biomarkers. (**A**) TNF-α, (**B**) IL-1β, (**C**) IL-6, (**D**) IL-10. Bars are mean ± SD. *n* = 8 rats/group. * *p* < 0.05, ** *p* < 0.01, vs. Control offspring (Con); ^#^
*p* < 0.05, vs. GDM offspring (GDM); ^$^
*p* < 0.05, ^$$^
*p* < 0.01, vs. n-3 Adequate-GDM offspring (n-3 Adq-GDM).

**Figure 5 nutrients-11-01699-f005:**
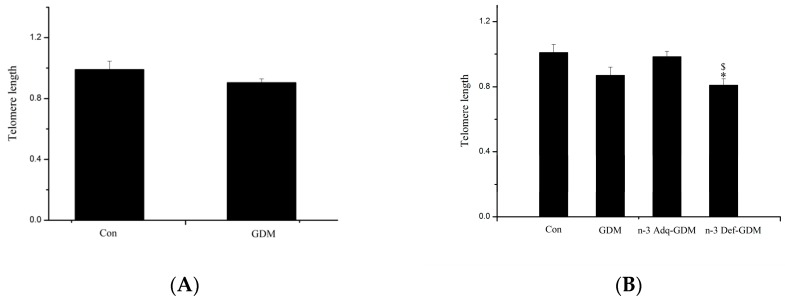
Influence of gestational diabetes mellitus (GDM) on telomere length (TL) of the liver of offspring and the effect of n-3 polyunsaturated fatty acids (*n*-3 PUFA) and *n*-6 PUFA on telomere length of the liver of GDM offspring. (**A**) At weaning; (**B**) at 11 months. *n* = 8 rats/group. * *p* < 0.05, vs. Control offspring (Con); ^$^
*p* < 0.05, vs. n-3 Adequate-GDM offspring (n-3 Adq-GDM).

**Figure 6 nutrients-11-01699-f006:**
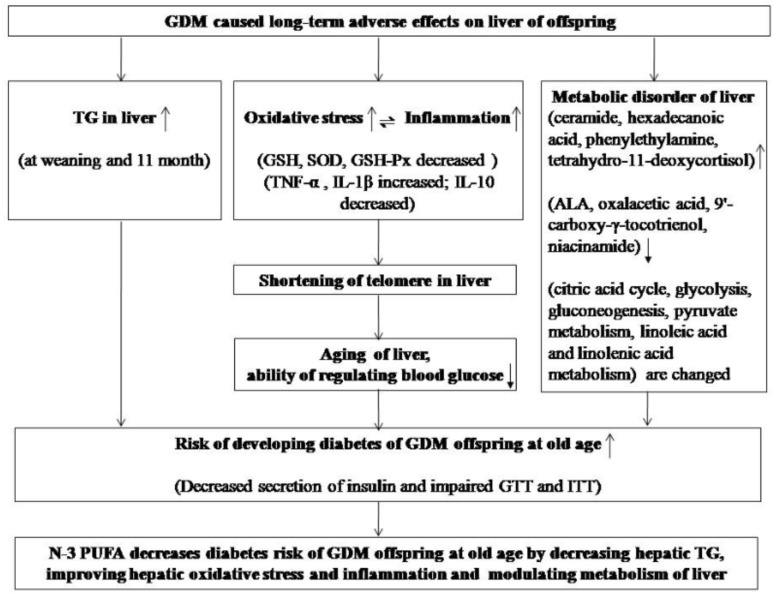
A summary figure showing all of the results.

**Figure 7 nutrients-11-01699-f007:**
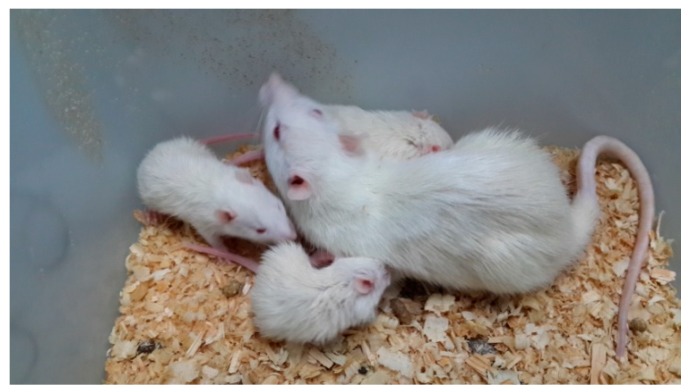
The gestational diabetes mellitus (GDM) offspring at weaning.

**Table 1 nutrients-11-01699-t001:** Biochemical index of gestational diabetes mellitus (GDM) offspring at weaning.

3 Week (Weaning)	Con	GDM
Fasting glucose (mmol/L)	6.17 ± 1.24	6.01 ± 1.3
Postprandial glucose (mmol/L)	7.34 ± 0.61	7.57 ± 0.49
Fasting insulin (mU/L)	12.01 ± 0.59	9.7 ± 0.46 *
Postprandial insulin (mU/L)	13.23 ± 0.78	20.17 ± 1.45 **
TG (mmol/L)	0.45 ± 0.1	0.46 ± 0.11
TC (mmol/L)	1.76 ± 0.19	1.98 ± 0.16

Data are presented as mean ± SD. *n* = 8–10 offspring rats/group. * *p* < 0.05, ** *p* < 0.01 vs. Control offspring (Con).

**Table 2 nutrients-11-01699-t002:** Biochemical index of GDM offspring at 11 months of age and n-3 polyunsaturated fatty acids (*n*-3 PUFA) effect.

11 Months old	Con	GDM	n-3 Adq-GDM	n-3 Def-GDM
Glucose (mmol/L)	6.05 ± 0.37	8.39 ± 2.07	6.3 ± 0.54	8.38 ± 1.66
Insulin (mU/L)	11.73 ± 0.55	8.03 ± 0.47 **	10 ± 0.46 *^##^	7.8 ± 0.78 **
TG (mmol/L)	0.79 ± 0.13	0.63 ± 0.19	0.43 ± 0.19 *^#^	0.61 ± 0.19
TC (mmol/L)	1.87 ± 0.04	1.79 ± 0.36	0.69 ± 0.16 **^##^	1.7 ± 0.17
HDL (mmol/L)	0.53 ± 0.03	0.52 ± 0.18	0.27 ± 0.04 **^##^	0.47 ± 0.03
TC/HDL	3.52 ± 0.12	3.44 ± 0.66	2.55 ± 0.28 **^##^	3.62 ± 0.19

Data are presented as mean ± SD. *n* = 8–10 offspring rats/group. * *p* < 0.05, ** *p* < 0.01, vs. Control offspring (Con); ^#^
*p* < 0.05, ^##^
*p* < 0.01, vs. GDM offspring (GDM).

**Table 3 nutrients-11-01699-t003:** Representative metabolites altered in the liver of gestational diabetes mellitus (GDM) offspring at 11 months of age and the effects of n-3 polyunsaturated fatty acids (*n*-3 PUFA) and *n*-6 PUFA on them.

Identification	RT (Min)	m/z	Changing Trend			Significance
			GDM vs. Con	n-3adq vs. GDM	n-3def vs. GDM	
Ceramide (d18:1/16:0)	17.78	560.5073	↑	↓ *	-	Biomarker for diabetes; impair insulin signaling and cause insulin resistance; increase oxidative stress; promote inflammation; contribute to non-alcohol fatty liver disease
Tetrahydro-11-deoxycortisol	11.56	337.2756	↑	↓ **	-	Impact cortisol and further impact insulin production and glucose metabolism; inhibit glycogen synthesis; cause insulin resistance
9’-Carboxy-γ-tocotrienol	16.50	395.2234	↓	-	↓ **	Antioxidant effect; improve glycemic control; prevent hyperlipidemia; suppress inflammation
α-Linolenic acid	11.93	570.3029	↓	-	↓ **	Decreases diabetic risk; improve insulin resistance; improve oxidative stress and inflammation; regulate lipid metabolism; improve non-alcohol fatty liver disease
Hexadecenoic acid	15.99	271.2649	↑	-	-	Induce endoplasmic reticulum stress and insulin resistance; lipotoxicity; enhance oxidative stress and inflammation; contribute to non-alcohol fatty liver disease
Niacinamide	0.95	123.0515	↓	-	-	Prevent diabetes; protect β cell; antioxidative role; anti-inflammatory effect
Oxalacetic acid	18.17	154.9962	↓	-	↓ *	Impact citric acid cycle and glucose and lipid metabolism; decrease of it indicate gluconeogenesis
Phenylethylamine	2.52	122.0956	↑	-	↑ **	Indicate possibility of hepatic damage and hepatic encephalopathy

The “↑” and “↓” arrows represent a significant increasing or decreasing trend of metabolites of GDM offspring. Green arrows show modulating effect on altered metabolites, and the red shows aggravating results. “-” means no significant change. * *p* < 0.05, ** *p* < 0.01, vs. GDM offspring.

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
