# Peer review of "n-3 Polyunsaturated Fatty Acids Decrease Long-Term Diabetic Risk of Offspring of Gestational Diabetes Rats by Postponing Shortening of Hepatic Telomeres and Modulating Liver Metabolism"

_nutrients, 2019, doi:10.3390/nu11071699_

Round 1
Reviewer 1 Report
This manuscript of Gau et al. Investigate the effect of omega 3 polyunsaturated fatty acids on GD in rodent model. The authors performed many experiments, and there is a huge amount of data that need to be properly used in order to see the wood for the trees.
I will provide general comments that may help the authors in presentation of their manuscript:
1-You should be rational in which data to present to suite your hypothesis. I feel, you did load of experiments and displayed too much unnecessary data. This makes it harder for the reader to follow.
2-Table 4 the author need to display a concise table outlining the top metabolite relevant to their hypothesis instead of displaying this long information.
3- Results section require to be completely rewritten. The authors kind of dumped the list of metabolites and started to explain about each one in isolation. You need to discuss only relevant information in a cohesive manner. Discussion is extremely long and isolated, I lost the track.... I suggest that authors require English proofreading step.
4-Figures: They need improvement overall in presentation. Again be selective in which data you need to display. Too much not always better!
5. Reference list. Again extremely long, you need to be selective.
Author Response
Please see the attachment, thanks!

Reviewer 2 Report
The authors need to be modify the manuscript as follows:
English language needs to be improved.
A flow diagram showing the protocol f the study can be given
The results give in Table 4 -can the authors out another column and show the significance of the each altered metabolite.
A summary table showing all results may be given for easy reading and understanding.
Is it possible that there are alterations in the muscle tissue similar to liver.
Why gut was not analysed similar to liver.
Why fatty acid analysis of plasma, liver was not done to know how much of the fed n-3 fatty acids got incorporated.
Author Response
Please see the attachment, thanks!

Round 2
Reviewer 1 Report
The presentation of the manuscript has substantially improved, I can now see the wood of the tree. However, there are still ongoing issues that need further revision.
Major Points
The authors quantified a range of inflammation TNF-α, IL-1β, IL-6 and IL-10) and oxidative stress markers. They presented data in figure 3 and figure 4. Hoverer, they failed to reflect on this in the discussion. The only mention that I could find is on the 3ed paragraph in discussion (line 1007) “Our present study showed that the liver of GDM offspring exhibited oxidative stress and inflammation after they grew to the long-term age of 11 months, which were important factors in inducing diabetic risk” This is a very general statement, you need to reflect on the changes in each markers and support your discussion with the appropriate literature about the role of inflammation (TNF-α, IL-1β, IL-6 and IL-10) and oxidative stress markers that you measure on diabetes development.
The authors report changes in metabolites profile and inflammation factors in liver tissue, how these changes translate into the plasma or serum changes? There are data on serum TG have you measure inflammation markers and oxidative stress markers in serum? This is more biologically relevant markers in term of diabetes onset and development.
Your decision on which data to include and exclude still not satisfactory. Body weight and hepatic triglycerides are two major factor that affect diabetes development. Data require to be presented (maybe supplementary) and discussed in light of your hypothesis.
Flow diagram (Figure1) is useful, but I cannot see any added value to figure 6 that outlines major findings in its current format (maybe need improvement if decided to keep). Moving figure 7(Figure S1) and the rest of the metabolites (Table S1) to the supplementary data is OK. However, I cannot see the point of presenting figure 8 (Figure S2) in the supplementary data, what value will bring considering that you presented the complete list of metabolites. Maybe based on the editorial policy of the journal, the complete list of metabolites will be better presented as supplementary data, while the discussion should remain on the selected relevant ones.
Displaying the figures need improvement. For example, figure 2, the y axis (glucose concentration) requires to have the same scale i.e. 0-20 for easy comparison of changes between different stages. All figures require sufficient information under legends to interpret the data, the reader should be able to understand the figure without the need to go back to the methods.
Statistical analysis (267): Fully describe under this section all statistical analysis. Part of the statistical analysis is displayed on line 235 under: Data Processing, Multivariate Statistical Analysis and Metabolites Identification
Minor points
Line 2: The word improve was wrongly used in the title, abstract, and introduction to describe the effect of PUFA. It was stated: N-3 polyunsaturated fatty acids “improves” long-term diabetic risk. I guess the authors mean to say decrease or attenuate not improve? The same word used on lines 19, 25, 37, and 304.
Line 41: Telomeres, which are termini of chromosomes and contain specific DNA sequences: delete and : Telomeres, which are termini of chromosomes contain specific DNA sequences
Line 114: It is not appropriated to use the word kill, I would suggest the authors use more appropriate term like the animals were sacrificed. This was repeated on lines:161, 174
Line 91: The phrase long-term age of 11 months needs to be revised to 11 months old throughout the manuscript (Line 449).
Line 214 omit of Liver by UPLC-QTOF-MS: the description is mentioned in the paragraph, it would be better to leave the title as: Metabolomics Analysis.
Line 216: Non-targeted metabolomics analysis was performed on the liver: this should be phrased correctly, basically on tissues extracted from the liver. Similarly, in line 201: TNF-alpha…. Levels in the liver (it should be in liver tissue).
Line 244: to identify chemical structure hypothesis? What the authors mean by that? Does it mean to identify chemical structure prediction of the identified metabolites?
Line 281: pancreatic injury is a very broad term, the authors reported change in blood insulin which is a function of the beta cells only.
Line 445: diabetic risk with month age? This does not make sense?
Line 688: long term effects of GDM on offspring were poorly understood (should be are not were)
Line 1027: Our study is the first to investigate the long-term influence of GDM on hepatic metabolism. Hepatic lipid or hepatic glucose metabolism, you need to specify, the term hepatic metabolism is not clear.
Typos:
Line 3: telomere not temomere
Line 163: centrifuged at 3000 r/min?
Line 233: you missed °C after source temperature.
Line505: you have extra O before in the liver.
Reviewer 2 Report
Authors have improved the manuscript.
I suggest that all the authors try to incorporate all the tables and figures given in their cover letter be incorporated into the main manuscript and integrate into it so that the final manuscript will be more appealing and comprehensive. This is especially tru of fatty acid profile analysis.
I am surprised at the muscle weakness shown by the non-N-3 fatty acid fed group of animals. This is important that need to be given in the main manuscript. This is an important observation. This muscle weakness observed implies that these animals are unable to perform adequate exercise that in turn can enhance insulin resistance (due to lack of exercise) and predispose them to develop DM. This ia ll the more reason as to why a through analysis of muscle tissue is needed. It is possible that the muscle tissue of these animals have less number of mitochondria, actin-myosin filaments are improperly organised, the fatty acid pattern will be different. Some of these possible effects can be highlighted in the discussion section so that it will form the basis of future studies.
